# An Empirical Study of Frame Selection for Text-to-Video Retrieval

**Mengxia Wu[1], Min Cao[1]\*, Yang Bai[1], Ziyin Zeng[1], Chen Chen[2],**
**Liqiang Nie[3], Min Zhang[1]**

[1] Soochow University, [2] Institute of Automation, Chinese Academy of Sciences,
[3] Harbin Institute of Technology, Shenzhen
{mxwuwumx}@stu.suda.edu.cn, {mcao}@suda.edu.cn

## Abstract

Text-to-video retrieval (TVR) aims to find the most relevant video in a large video gallery given a query text. The intricate and abundant context of the video challenges the performance and efficiency of TVR. To handle the serialized video contexts, existing methods typically select a subset of frames within a video to represent the video content for TVR. How to select the most representative frames is a crucial issue, whereby the selected frames are required to not only retain the semantic information of the video but also promote retrieval efficiency by excluding temporally redundant frames. In this paper, we make the first empirical study of frame selection for TVR. We systemically classify existing frame selection methods into text-free and text-guided ones, under which we detailedly analyze six different frame selections in terms of effectiveness and efficiency. Among them, two frame selections are first developed in this paper. According to the comprehensive analysis on multiple TVR benchmarks, we empirically conclude that the TVR with proper frame selections can significantly improve the retrieval efficiency without sacrificing the retrieval performance.

## 1 Introduction

Text-to-video retrieval (TVR) aims to search for the most relevant video given a query text, involving intra-modal understanding and inter-modal alignment. Videos are composed of a sequence of frames, each of which can be considered as the primary processing unit for video understanding and alignment with the text. However, due to the constraints of computing resources, it is impractical to process all frames of one video for TVR. Therefore, existing methods (Liu et al., 2019; Gabeur et al., 2020; Lei et al., 2021; Luo et al., 2022) select a subset of frames from the original videos to perform TVR. Generally speaking, selecting fewer frames

---

\*Corresponding author

can benefit higher computational efficiency, while leading to the loss of visual details and thus impairing retrieval performance. Considering both effectiveness and efficiency, it is imperative to adopt an optimal frame selection method for TVR.

Current frame selection methods can be divided into text-free frame selection and text-guided frame selection. The text-free frame selection only explores the video contents themselves while ignoring their relevancy to the texts in selecting frames. Specifically, the conventional techniques (Chang et al., 1999; Divakaran et al., 2002; Chang, 2003) for frame selection primarily rely on the color features of frames. However, these approaches fall short of comprehending the semantic content of videos and are not readily applicable to end-to-end training in various video-related tasks. For simplicity, existing works (Lei et al., 2021; Luo et al., 2022; Li et al., 2022a) typically employ uniform frame selection (Uni) or random frame selection (Rand) to eliminate the temporally redundant frames that contribute little to the retrieval. In contrast, the text-guided frame selection is aimed at utilizing textual information to filter the inessential frames. We partition the text-guided frame selection into two categories: non-interactive frame selection (N-InT) (Gorti et al., 2022; Han et al., 2022) and interactive frame selection (InT) (Buch et al., 2022; Yang et al., 2023; Wang et al., 2022; Lin et al., 2022). The former directly calculates the similarities between the representations of frames and the text from two unimodal encoders without interaction, and then selects the frames with high similarities as the representatives of the video. The latter constructs a cross-modal interactive module with extra parameters to assess the relevancy between frames and the text.

Although persistent efforts have been made in frame selection, there still exist three main problems. 1) The critical clues to the retrieval in a video typically exhibit a non-uniform distribution. The

text-free frame selection with the Uni/Rand does not follow this distribution, which may filter out the key frames that carry the critical clues and eventually hamper the performance of TVR. And the text-guided frame selection circumvents this problem with the aid of text information, but at the expense of simpleness. Nevertheless, both of them have not considered the distribution of the video content itself. 2) The videos usually contain low-quality frames that are out-of-interest and less meaningful to TVR, leading to increased noise and decreased efficiency. These frames could caused by transitions, out-of-focus shots, inappropriate angles, etc. Yet, such frames are overlooked by the current text-free frame selection methods. 3) Currently, there is a noticeable absence of a comprehensive comparison and analysis of frame selection methods that take into account the aforementioned two factors. As a result, the optimal frame selection method for TVR still remains elusive, with no definitive answer yet.

To this end, we extend the current frame selection methods and conduct a thorough empirical study of the frame selection methods to explore the optimal one in this paper. Specifically, beyond the existing frame selections, we first develop a redundancy-aware frame selection (Redun-A) method for the first problem and a low-quality-aware frame selection method (LQ-A) for the second problem. Redun-A selects the most representative and diverse frames by clustering the video frames, and LQ-A chooses the frames of high quality by designing a scorer network and retaining frames with high scores, both of which belong to the text-free frame selection. Then, based on the existing frame selections and the proposed ones, we investigate the effect of each frame selection method on TVR, as well as the potential combinations of them. Through extensive study, we conclude that: (i) giving priority to the text-free frame selection, combining the proposed Redun-A and LQ-A is an optimal policy; (ii) by further considering the text-guided frame selection, the better trade-off between effectiveness and efficiency is achieved by the combination of Redun-A and N-InT.

Our main contributions can be summarized as follows:

- We present an empirical study of frame selection for TVR, encompassing four text-free frame selection methods and two text-based frame selection methods. Through a comprehensive analysis, we aim to explore the optimal frame selection policy that balances effectiveness and efficiency.

- We introduce two text-free frame selection methods, i.e., the redundancy-aware and low-quality-aware frame selection, thus providing broader views for the frame selection.

- We conduct a comprehensive evaluation of the frame selection methods on the three common benchmarks, i.e., MSR-VTT (Xu et al., 2016), DiDemo (Anne Hendricks et al., 2017) and ActivityNet Captions (Caba Heilbron et al., 2015). A significant trade-off between performance and efficiency on the benchmarks can be achieved by the explored optimal frame selection policy.

## 2 Related Work

### 2.1 Text-to-Video Retrieval

The early TVR methods (Yu et al., 2018; Liu et al., 2019; Gabeur et al., 2020) focused on designing a fusion module for cross-modal learning from the offline extracted video and text features. In recent years, visual-language pre-training technologies have begun to push the development of TVR dominantly. CLIP4Clip (Luo et al., 2022) conducted an empirical study on video aggregation schemes for transferring the CLIP (Radford et al., 2021), which has been pre-trained on large-scale text-image pairs, to TVR. This study demonstrated impressive performance in TVR with the aid of CLIP. As a result, a series of works have focused on excavating the power of CLIP by designing various alignment strategies (Ma et al., 2022; Fang et al., 2023; Gorti et al., 2022; Chen et al., 2023) for performance improvement or plugging token selection strategies (Zhao et al., 2022; Liu et al., 2022) for efficiency improvement. CenterCLIP (Zhao et al., 2022) performed token selection by clustering numerous video patch tokens within each short video segment. However, this method limited the selection to the local content of the video and resulted in inefficient information selection through token clustering. Beyond adopting CLIP, a two-stream network without the fusion module, as TVR backbone, LiteVL (Chen et al., 2022) constructed a BLIP-based (Li et al., 2022b) model, which incorporates a pre-trained deep fusion module into TVR, and exhibits better performance but lower

efficiency than the CLIP-based works. Differently, this paper makes an empirical study of frame selection and explores an optimal frame selection policy for TVR, aiming at achieving the trade-off between retrieval performance and efficiency.

## 2.2 Frame Selection for TVR

Frame selection is an important process for TVR. It can filter out the inessential frames in the video for a high-effectiveness and high-performance TVR. Uniform and random frame selections are two common frame selection methods, by which some works (Luo et al., 2022; Lei et al., 2021) empirically explored the proper frame rate for TVR to balance efficiency and effectiveness. However, due to no reasonable guidance for frame selection, both methods are prone to lose some key frames beneficial to TVR, resulting in a limited trade-off between efficiency and effectiveness. Actually, texts are the natural guidance for frame selection in TVR. Directly calculating the similarities between representations of the frame and text is one of the ways (Gorti et al., 2022; Han et al., 2022). Beyond that, TW-BERT and FinCo (Yang et al., 2023; Wang et al., 2022) proposed to simply concatenate the representations of frames with the text, and the concatenated representations were then input to a scorer network, which comprised a fully connected layer followed by a softmax layer, to assign a score to each frame. The resulting scores are adopted to guide the frame selection. To better fuse the inter-modal information during frame selection, some works (Buch et al., 2022; Lin et al., 2022) constructed a self-attention-based encoder to fuse the frames and text representations. Different from the above text-guided frame selection methods, we employ a cross-attention-based encoder to fuse the frames and text representations, and then select the textual-relevant frames based on the attention weights in InT frame selection. Moreover, this paper provides a comprehensive analysis of the frame selections to figure out the optimal one for TVR.

## 3 Methods

In this section, we first introduce the model architecture used for the empirical study of frame selection. Then, we elaborate on the core content of our empirical study, including four text-free frame selection methods (i.e., uniform, random, redundancy-aware, and low-quality-aware frame selections) and two text-guided frame selection methods (i.e., non-interactive and interactive frame selections). Finally, we present the training and inference processes.

### 3.1 Model Architecture

Following BLIP (Li et al., 2022b), the model architecture comprises three main components: a vision encoder, a text encoder, and a multimodal encoder, as shown in Figure 1 (a). Details for each component are as follows.

**Vision Encoder.** We employ the ViT-B/16 (Dosovitskiy et al., 2020) as our vision encoder. Given a video $V$, following the general practice in TVR methods (Lei et al., 2021; Luo et al., 2022), we first uniformly pre-sample $N$ frames $\mathcal{V} = \{F_1, F_2, \cdots, F_N\}$ from the video to represent video contents. Then we conduct frame selection based on $\mathcal{V}$ to improve the retrieval efficiency. For the uniform and random frame selections, we directly operate on $\mathcal{V}$ and obtain $K$ $(K < N)$ frames $\tilde{\mathcal{V}} = \{F_{s_1}, F_{s_2}, \cdots, F_{s_K}\}$, where $s_k \in \{1, 2, \ldots, N\}$ denotes the index of the selected frame. Then, each frame $F_{s_k}$ is fed into ViT-B/16 and encoded as a sequence of representations $f_{s_k} = \{f_{s_k}^{cls}, f_{s_k}^1, \cdots, f_{s_k}^P\}$, where $P$ is the number of the frame patches and $f_{s_k}^{cls}$ denotes the representation of [CLS] token and can be viewed as the global representation of $F_{s_k}$. For the other four frame selections (i.e., redundancy-aware, low-quality-aware, non-interactive, and interactive frame selections), we first encode the frames in $\mathcal{V}$ using ViT-B/16, based on which we select $K$ key frames. To sum up, the uniform and random frame selections are performed in image frame space, while the redundancy-aware, low-quality-aware, non-interactive, and interactive frame selections are carried out in feature representation space. These frame selection methods will be detailed in Section 3.2.

To this end, we can obtain the representations of the selected $K$ frames. Based on this, we compute the global representation of the video $f_V$. We first map the [CLS] representations $f_{s_k}^{cls}$ of all $K$ frames to a lower dimension (256-d) by a linear transformation, denoted as $f_{s_k}^{cls\prime}$. For the uniform, random and redundancy-aware frame selections, we conduct mean-pooling on the $f_{s_k}^{cls\prime}$ for generating the global representation of the video:

$$f_V = \frac{1}{K} \sum_{k=1}^{K} f_{s_k}^{cls\prime}. \tag{1}$$

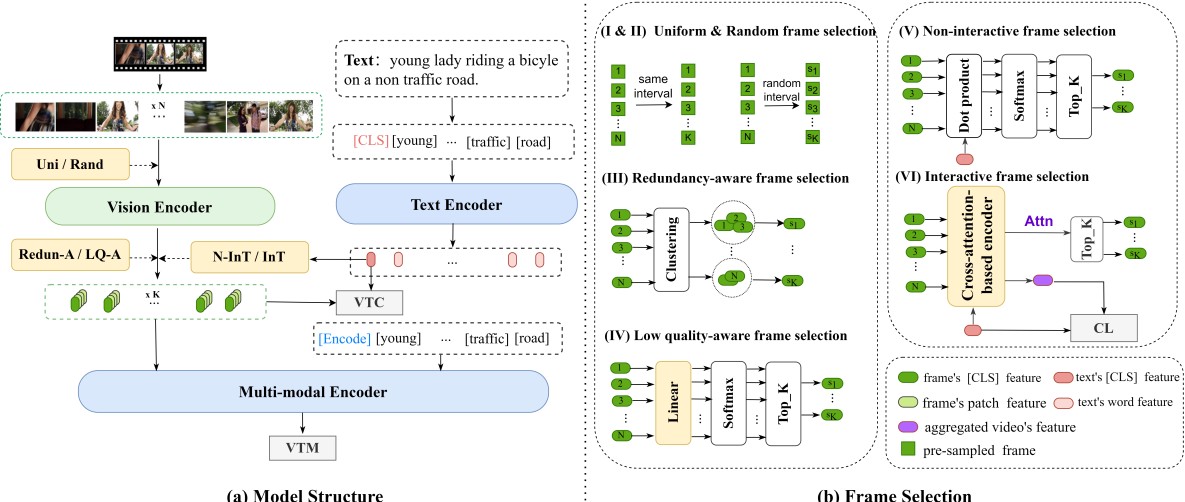

**(a) Model Structure**

**(b) Frame Selection**

Figure 1: An illustration of the model architecture and frame selection methods. (a) The model architecture consists of three encoders, i.e., vision, text, and multimodal encoders. The uniform and random frame selections work in image frame space, while the other four in feature space. (b) We group all the frame selection methods into two types, i.e., text-free frame selection (I ~IV) and text-guided frame selection (V ~VI). $\{s_1, ..., s_K\}$ denotes the sequence of key frames. The low-quality-aware and interactive frame selection methods bring in extra parameters, while the others are parameter-free.

While for the other four frame selection methods, we obtain $f_V$ by a weighted average of $f_{s_k}^{cls\prime}$:

$$f_V = \sum_{k=1}^{K} softmax(\alpha_{s_k}) f_{s_k}^{cls\prime}, \qquad (2)$$

where $\alpha_{s_k}$ is the score of the frame $F_{s_k}$ computed by the corresponding frame selection method.

**Text Encoder.** We apply BERT-base (Devlin et al., 2018) as the text encoder that transforms an input text $T$ with $M$ tokens into a sequence of representations $\{t^{cls}, t^1, \cdots t^M\}$, where the $t^{cls}$ is the representation of the [CLS] token. We also map $t^{cls}$ to the lower dimension (256-d) as the global representation of the text, denoted as $f_T$.

**Multimodal Encoder.** We adopt the BERT-base together with a cross-attention layer inserted in each transformer block as the multimodal encoder. The BERT-base shares the parameters with the text encoder. The representations of the selected K frames from the vision encoder act as the visual inputs of the multimodal encoder. For the textual inputs, following BLIP (Li et al., 2022b), we append an [Encode] token to $T$, and its encoded representation $t^{enc}$ is denoted as the multimodal representation of the video-text pair. In the cross-attention layer, we take textual inputs as the query and visual inputs as the key and value.

## 3.2 Frame Selection

According to the participation of text, we categorize the frame selection methods into two groups:

the text-free and text-guided frame selections, as shown in Figure 1 (b). In the following, we detail each frame selection method.

### 3.2.1 Text-Free Frame Selection

The text-free frame selection chooses frames only from the perspective of video contents themselves, including the uniform, random, redundancy-aware, and low-quality-aware frame selection methods. The former two frame selection methods are performed on the image frame level, while the latter two are on the feature representation level.

**Uniform/Random Frame Selection.** The uniform and random frame selections are the most straightforward frame selection methods. Given the pre-sampled frames $\mathcal{V}$, the uniform frame selection evenly selects $K$ frames with the same intervals from $\mathcal{V}$, while the random sampling selects with random intervals. The selected frames are then sent to the vision encoder for frame representation.

**Redundancy-aware Frame Selection.** The uniform or random frame selection neglects the irregular distribution of crucial information within video contents, leading to the tendency to lose the crucial frames. For this, we develop a redundancy-aware frame selection, which is dedicated to reserving the most representative and diverse frames from each video. Specifically, given a sequence of lower-dimensional [CLS] representations of each frame in $\mathcal{V}$, denoted as $f^{cls\prime} = \{f_1^{cls\prime}, f_2^{cls\prime}, \cdots, f_N^{cls\prime}\}$, we utilize

k-medoids++ clustering algorithm (Zhao et al., 2022) to partition them into $K$ groups. The frames in the same group typically exhibit similar video contents, indicating redundancy within the video. The center frame of each group is considered the most representative one. Naturally, we select the frame whose [CLS] representation corresponds to the center of each group.

**Low-quality-aware Frame Selection.** The presence of low-quality frames in videos, which are generally out of interest, has a negative impact on both retrieval performance and efficiency. Nonetheless, the low-quality frames are ignored by the existing text-free frame selection methods. In this situation, we propose a low-quality-aware frame selection to filter them. In detail, we append a scorer network after the vision encoder to assess the quality score $\alpha_i$ of each frame:

$$\alpha_i = softmax(FC(f_i^{cls\prime})), \qquad (3)$$

where $FC(\cdot)$ denotes the fully connected layer that map the $f_i^{cls\prime}$ $(i = 1, 2, \cdots, N)$ to one dimension. Given the absence of explicit annotations indicating frame quality, we instead leverage the paired text as weak supervision to optimize the scorer network. This strategy is based on the assumption that frames highly relevant to the text are less likely to be of low quality. Specifically, during the training process, we employ the scores of selected frames as weights to aggregate the frames features as Eq. 2, and then utilize the VTC loss (detailed in section 3.3.1) to optimize the scorer network. Finally, we retain the frames with the top $K$ scores. During inference, we score each video frame by the trained scorer network to filter out the low-quality frames.

### 3.2.2 Text-guided Frame Selection

In contrast to the text-free frame selection, the text-guided frame selection considers the query text information to help filter the inessential frames, which contain the non-interactive and interactive frame selections.

**Non-interactive Frame Selection.** It is efficient to directly calculate the similarities between the representations of frames and the given text for the frame selection. Specifically, we compute the cosine distances, followed by a softmax layer, as the relevancy score of frame $F_i$ and the given text $T$, denoted as $\alpha_i$:

$$\alpha_i = softmax(s(f_i^{cls\prime}, f_T)), \qquad (4)$$

where $s(\cdot, \cdot)$ denotes the cosine similarity. The frames with the highest $K$ similarities to the given text are selected.

**Interactive Frame Selection.** The interactive frame selection constructs a specific multimodal-based selection module to model the relevancy between frames and the given text. In our implementation, we design a cross-attention-based encoder for frame selection, which consists of a single transformer layer. Specifically, in the cross-attention module, the text global representation $f_T$ is projected into a query $Q_t$, and the frames' [CLS] representations $f_i^{cls\prime}$ $(i = 1, \cdots, N)$ are projected into key $K_f$ and value $V_f$. We then adopt the scaled dot product attention to aggregate these representations, denoted as $f_{T\prime}$:

$$f_{T\prime} = Attn(Q_t, K_f)V_f, \qquad (5)$$

$$Attn(Q_t, K_f) = softmax(\frac{Q_t K_f^\top}{\sqrt{d}}), \quad (6)$$

where $d$ is the dimension of $K_f$. The dot product attention $Attn(Q_t, K_f) \in \mathcal{R}^{1 \times N}$ provides relevancy weights from a text to each frame. Through the attention mechanism, the fused video feature $f_{T\prime}$ is encouraged to incorporate the text semantics. To optimize the selection module, we employ the Contrastive Loss (CL) to pull closer the $f_{T\prime}$ and the paired $f_T$ in each batch $B$, denoted as $\mathcal{L}_{cl}$:

$$\mathcal{L}_{cl} = -\frac{1}{B} \sum_i^B \log \frac{e^{s(f_T^i, f_{T\prime}^i)/\lambda}}{\sum_{j=1}^B e^{s(f_T^j, f_{T\prime}^i)/\lambda}}, \quad (7)$$

where $\lambda$ is a learnable temperature parameter. Finally, we select the frames with the top $K$ attention scores to the text.

## 3.3 Training and Inference

### 3.3.1 Training

During training, we optimize the model mainly with two losses, i.e., video-text contrastive loss (VTC) and video-text matching loss (VTM). In particular, we also append the Contrastive Loss (CL) to optimize the selection encoder of the interactive frame selection.

**Video-Text Contrastive (VTC)** pulls the representations of paired video and text close and pushes away the unpaired ones. Specially, given the [CLS] representation of video $f_V$ and text $f_T$, we calculate in-batch symmetric VTC loss as follows:

$$\mathcal{L}_{vtc} = \frac{1}{2}(\mathcal{L}_{t2v} + \mathcal{L}_{v2t}), \qquad (8)$$

Table 1: Performance comparison of different frame selection methods on MSR-VTT and DiDeMo. We gray out the best performance in each row. The 'base' in the Frame column represents the video processing only with the pre-sampling and without further using the frame selection.

| Dataset | Frame | Text-Free | | | | | | | | Text-Guided | | | |
|---|---|---|---|---|---|---|---|---|---|---|---|---|---|
| | | Uni | | Rand | | Redun-A | | LQ-A | | N-InT | | InT | |
| | | R@1↑ | R@sum↑ | R@1↑ | R@sum↑ | R@1↑ | R@sum↑ | R@1↑ | R@sum↑ | R@1↑ | R@sum↑ | R@1↑ | R@sum↑ |
| MSR-VTT | 16(base) | 52.9 | 212.9 | 52.9 | 212.9 | 52.9 | 212.9 | 52.9 | 212.9 | 52.9 | 212.9 | 52.9 | 212.9 |
| | 16⇒12 | 51.7 | 212.6 | 51.8 | 212.8 | 52.3 | 213.0 | 51.1 | 209.6 | 50.4 | 210.6 | 50.8 | 207.7 |
| | 16⇒8 | 50.9 | 209.9 | 52.4 | 211.1 | 50.8 | 210.5 | 49.6 | 206.5 | 47.3 | 202.1 | 48.9 | 204.3 |
| | 16⇒6 | 51.4 | 209 | 49.7 | 206.9 | 52.2 | 211.4 | 47.8 | 202.5 | 46.3 | 200.3 | 48.5 | 204.8 |
| | 16⇒4 | 47.9 | 202.0 | 47.8 | 201.9 | 49.0 | 205.3 | 45.2 | 196.6 | 44 | 193.8 | 46.0 | 197.4 |
| | 16⇒1 | 24.3 | 125.6 | 26.5 | 136.3 | 39.4 | 176.1 | 33.8 | 158 | 41.7 | 185.4 | 32.6 | 156.3 |
| DiDeMo | 32(base) | 58.4 | 229.2 | 58.4 | 229.2 | 58.4 | 229.2 | 58.4 | 229.2 | 58.4 | 229.2 | 58.4 | 229.2 |
| | 32⇒24 | 57.8 | 228.1 | 58.7 | 228.6 | 58.3 | 229.7 | 59.2 | 230.7 | 58.6 | 228.6 | 58.6 | 226.4 |
| | 32⇒16 | 58.1 | 227.4 | 58.0 | 227.6 | 59.3 | 231.1 | 57.7 | 228.7 | 56.3 | 223.9 | 57.6 | 225.6 |
| | 32⇒12 | 56.9 | 225.7 | 57.6 | 226.5 | 58.3 | 231.3 | 56.6 | 226.7 | 55.2 | 224.2 | 55.7 | 222.8 |
| | 32⇒8 | 54.6 | 223.4 | 53.8 | 222.9 | 57.7 | 230.7 | 55.9 | 224.5 | 54.0 | 221.2 | 55.2 | 220.9 |
| | 32⇒4 | 52 | 215 | 52.5 | 215.4 | 57.0 | 225.8 | 51.2 | 213.5 | 51.6 | 217.2 | 49.8 | 215.5 |
| | 32⇒1 | 33.2 | 166.3 | 34.2 | 164.5 | 38.5 | 180.6 | 36.8 | 173.5 | 47.2 | 203.6 | 41.7 | 191.8 |

where $\mathcal{L}_{t2v}$ and $\mathcal{L}_{v2t}$ denotes text-to-video and video-to-text loss, respectively, formulated as:

$$\mathcal{L}_{t2v} = -\frac{1}{B}\sum_{i}^{B}\log\frac{e^{s(f_V^i, f_T^i)/\tau}}{\sum_{j=1}^{B}e^{s(f_V^j, f_T^i)/\tau}}, \quad (9)$$

$$\mathcal{L}_{v2t} = -\frac{1}{B}\sum_{i}^{B}\log\frac{e^{s(f_V^i, f_T^i)/\tau}}{\sum_{j=1}^{B}e^{s(f_V^i, f_T^j)/\tau}}, \quad (10)$$

where $\tau$ is a learnable temperature parameter and $B$ is the batch size.

**Video-Text Matching (VTM)** determines whether a video-text pair is matched or not. We feed the multimodal encoder's output $t^{enc}$ into a binary classifier to predict a two-class probability $p_{vtm}$. The VTM loss is formulated as:

$$\mathcal{L}_{vtm} = \mathbb{E}_{(V,T)\sim D}H(y_{vtm}, p_{vtm}), \quad (11)$$

where $y_{vtm}$ is a 2-dimensional one-hot vector representing the ground-truth label.

In conclusion, the full training objective is:

$$\mathcal{L} = \mathcal{L}_{vtm} + \mathcal{L}_{vtc} + \beta\mathcal{L}_{cl}, \quad (12)$$

where $\beta = 1$ when we use the interactive frame selection, otherwise $\beta = 0$.

### 3.3.2 Inference

To speed up the inference, as BLIP (Li et al., 2022b), we first compute the feature similarities $S_{t2v}$ for all text-video pairs by the vision and text encoders, and then choose the first top-128 video candidates according to $S_{t2v}$ for further ranking by the multimodal encoder. In addition, with our frame selections, the inference is more efficient.

## 4 Experiment

### 4.1 Experimental Settings

**Datasets and Implementation.** We conduct experiments on three common TVR benchmarks, including MSR-VTT (Xu et al., 2016), DiDeMo (Anne Hendricks et al., 2017) and ActivityNet Captions (Caba Heilbron et al., 2015). The details of each benchmark and implementation are introduced in Appendix A and B, respectively.

**Evaluation Metric.** To evaluate the retrieval performance, we employ three retrieval metrics: recall at rank M (R@M, M=1, 5, 10), median rank (MdR) and the sum of the recall (R@sum). R@M calculates the percentage of the successful retrieval in which the paired sample to the query is found in the top-M retrieved results. MdR measures the median rank of correct items in the retrieved ranking list. R@sum is calculated by summing R@1, R@5, and R@10, reflecting the overall retrieval performance.

### 4.2 Analysis of Frame Selection

In this section, we first compare the frame selection methods on retrieval performance. Subsequently, we investigate the impact of each frame selection method on retrieval efficiency. At last, based on the performance and efficiency analysis, we explore combining multiple frame selection methods to determine the optimal method. All the analyses are based on the experimental results on MSR-VTT and DiDeMo, representing two canonical TVR types, i.e., standard sentence-to-video retrieval and

Table 2: Efficiency comparison of frame selection methods under the best performance. MeM is the max GPU memory cost during training. Time is the interaction duration per query with all videos during inference.

| Method | MSRVTT | | | | | DiDeMo | | | | |
|---|---|---|---|---|---|---|---|---|---|---|
| | Frame | R@1 | MeM(GB) | GLOPs | Time(ms) | Frame | R@1 | MeM(GB) | GLOPs | Time(ms) |
| Base | 16 | 52.9 | 19.2 | 1107.6 | 245.8 | 32 | 58.4 | 33.4 | 1997.1 | 462.9 |
| Uni | 16⇒12 | 51.7 | 15.8 | 836.7 | 191.9 | 32⇒24 | 57.8 | 26.4 | 1345.1 | 359.4 |
| Rand | 16⇒12 | 51.8 | 15.8 | 836.7 | 191.9 | 32⇒24 | 58.7 | 26.4 | 1345.1 | 359.4 |
| Redun-A | 16⇒6 | 52.2 | 14.9 | 861.7 | 112.1 | 32⇒16 | 59.3 | 26.4 | 1564.1 | 241.9 |
| LQ-A | 16⇒12 | 51.1 | 17.6 | 1042.2 | 189.8 | 32⇒24 | 59.2 | 30.0 | 1842.8 | 359.9 |
| N-InT | 16⇒12 | 50.4 | 17.6 | 1070.1 | 197.2 | 32⇒24 | 58.6 | 30.0 | 1854.6 | 365.7 |
| InT | 16⇒12 | 50.8 | 17.6 | 1098.6 | 210.6 | 32⇒24 | 58.6 | 30.0 | 1895.7 | 410.1 |

Table 3: Performance of combinations of frame selection methods. The best results are **bold**. The second best results are underline.

| Redun-A | LQ-A | N-InT | MSR-VTT | | | DiDeMo | | |
|---|---|---|---|---|---|---|---|---|
| | | | Frame | R@1↑ | R@sum↑ | Frame | R@1↑ | R@sum↑ |
| | | | 16 | **52.9** | 212.9 | 32 | 58.4 | 229.2 |
| ✓ | | | 16⇒12 | 52.3 | 213.0 | 32⇒16 | 59.3 | 231.1 |
| | ✓ | | 16⇒12 | 51.1 | 209.6 | 32⇒24 | 59.2 | 230.7 |
| | | ✓ | 16⇒12 | 50.4 | 210.6 | 32⇒24 | 58.6 | 228.6 |
| ✓ | ✓ | | 16⇒12 | 52.7 | 213.2 | 32⇒16 | 59.5 | 232.1 |
| ✓ | | ✓ | 16⇒12 | 52.6 | **213.9** | 32⇒16 | **59.7** | 232.2 |
| ✓ | ✓ | ✓ | 16⇒12 | 52.6 | 213.1 | 32⇒16 | 59.2 | **232.3** |

paragraph-to-video retrieval, respectively.

### 4.2.1 Effectiveness of Each Frame Selection

Frame selection aims to select fewer $K$ frames to represent a video for retrieval. Table 1 shows the retrieval performance under different frame selection methods with the setting of $K \in \{12, 8, 6, 4, 1\}$ for the MSR-VTT and $K \in \{24, 16, 12, 8, 4, 1\}$ for the DiDeMo. We can see that: (1) Redun-A exhibits an overall superiority compared to the existing methods (i.e., Uni and Rand) in reducing temporal redundancy in videos. Compared to the widely-used Uni and Rand which have not considered the distribution of crucial information in the video contents, Redun-A clusters the frames with similar contents to select the most representative ones and discard other redundant frames. As a result, Redun-A can effectively mitigate the performance decrease caused by the reduction of frame number. (2) When only selecting a single frame (i.e., $K = 1$) for TVR to maximize the retrieval efficiency, the text-guided frame selections, especially the N-InT, present a significant advantage compared to the text-free frame selections. The text-guided frame selection can retain the most query-relevant frame in a video and facilitate the retrieval. (3) When considering more frames (i.e., $K > 1$) for higher performance, the text-free frame selec-

tion performs better than the text-guided frame selection. We conjecture that the text-guided frame selection introduces the textual bias for TVR. This means that by emphasizing the text-related frames, the similarities of videos partially related to the given text may be improved, potentially resulting in extra disturbance for TVR.

### 4.2.2 Efficiency of Each Frame Selection

In this subsection, we explore the retrieval efficiency of different frame selection methods under the best performance in Table 2. Uni and Rand determine the $K$ key frames prior to the vision encoder, which means only fewer $K$ frames need to be encoded, resulting in the lowest memory cost and GLOPs. The Redun-A filters out most of the redundant frames through clustering and only retains 16 most representative frames, which gives rise to the fastest inference speed, simultaneously accompanied with a remarkable performance of 59.3% at R@1. Although the LQ-A achieves competitive performance compared to Redun-A, it comes at the cost of increased frame count, resulting in lower efficiency. The N-InT simply uses a dot product between the whole frames and the given query to select the key frames, bringing a marginal additional time consumption and comparable efficiency. Since the InT requires an additional fusion for the selection, it eventually exhibits an inferiority of retrieval efficiency. In conclusion, the Redun-A and N-InT are two reasonable individual frame selection methods for text-free and text-guided frame selection, respectively.

### 4.2.3 Combinations of Frame Selections

We combine various frame selections to explore a better trade-off between efficiency and performance. According to the aforementioned analysis of each individual frame selection method, we choose Redun-A for the text-free frame selection

Table 4: Results of text-video retrieval on MSR-VTT and DiDeMo datasets. The numbers on the left and right of ⇒ respectively denote the number of frames sampled from raw videos and frames used for inter-modal interaction after frame selection. † denotes our implementation of baselines.

| Method | MSR-VTT | | | | | | DiDeMo | | | | | |
|---|---|---|---|---|---|---|---|---|---|---|---|---|
| | Frames | R@1↑ | R@5↑ | R@10↑ | R@sum↑ | MdR↓ | Frames | R@1↑ | R@5↑ | R@10↑ | R@sum↑ | MdR↓ |
| CE (Liu et al., 2019) | - | 20.9 | 48.8 | 62.4 | 132.1 | 6 | - | 16.1 | 41.1 | 82.7 | 139.9 | 8.3 |
| ClipBERT (Lei et al., 2021) | 16 | 22.0 | 46.8 | 59.9 | 128.7 | 6 | 16 | 20.4 | 48.0 | 60.8 | 129.2 | 6 |
| Frozen (Bain et al., 2021) | 4 | 32.5 | 61.5 | 71.2 | 165.2 | 3 | 4 | 34.6 | 65.0 | 74.7 | 174.3 | 3 |
| TW-BERT (Yang et al., 2023) | 20⇒8 | 38.4 | 65.1 | 76.6 | 180.1 | 3 | 20⇒8 | 41.8 | 71.1 | 81.2 | 194.1 | 4 |
| CLIP4Clip (Luo et al., 2022) | 12 | 44.5 | 71.4 | 81.6 | 197.5 | 2 | 64 | 43.4 | 69.9 | 80.2 | 193.5 | 2 |
| MOF (Han et al., 2022) | 12⇒4 | 40.5 | 68.2 | 79.5 | 188.2 | 2 | 12⇒4 | 41.3 | 68.5 | 79.4 | 189.2 | 2 |
| CAMoE (Cheng et al., 2021) | 16 | 47.3 | 74.2 | 84.5 | 206 | 2 | - | - | - | - | - | - |
| CenterCLIP (Zhao et al., 2022) | 12 | 48.4 | 73.8 | 82.0 | 204.2 | 2 | - | - | - | - | - | - |
| X-CLIP (Ma et al., 2022) | 12 | 49.3 | 75.8 | 84.8 | 209.9 | 2 | 64 | 47.8 | 79.3 | - | - | - |
| Ts2net (Liu et al., 2022) | 12 | 49.4 | 75.6 | 85.3 | 210.3 | 2 | 64 | 41.8 | 71.6 | 82.0 | 195.4 | 2 |
| TABLE (Chen et al., 2023) | 12 | 47.1 | 74.3 | 82.9 | 204.3 | 2 | 32 | 47.9 | 74.0 | 82.1 | 204 | 2 |
| HBI (Jin et al., 2023) | 12 | 48.6 | 74.6 | 83.4 | 206.6 | 2 | 64 | 46.9 | 74.9 | 82.7 | 204.5 | 2 |
| Cap4Video (Wu et al., 2022) | 12 | 51.4 | 75.7 | 83.9 | 211 | 2 | 64 | 52.0 | 79.4 | 87.5 | 218.9 | 1 |
| X-pool† (Gorti et al., 2022) | 12 | 46.6 | 73 | 82.9 | 202.5 | 2 | 32 | 44.7 | 72.4 | 80.5 | 197.6 | 2 |
| X-pool(Redun-A+LQ-A) | 12⇒8 | 46.2 | 72.7 | 82.5 | 201.4 | 2 | 32⇒16 | 44.9 | 73.1 | 82.0 | 200.0 | 2 |
| X-pool(Redun-A+N-InT) | 12⇒8 | 46.7 | 72.6 | 83.2 | 202.5 | 2 | 32⇒16 | 45.2 | 73.1 | 82.5 | 200.8 | 2 |
| BLIP† (Li et al., 2022b) | 16 | **52.9** | 75.9 | 84.1 | 212.9 | 1 | 32 | 58.4 | 82.8 | 88.0 | 229.2 | 1 |
| BLIP(Redun-A+LQ-A) | 16⇒12 | 52.7 | 75.9 | 84.6 | 213.2 | 1 | 32⇒16 | 59.5 | 83.5 | **89.1** | 232.1 | 1 |
| BLIP(Redun-A+N-InT) | 16⇒12 | 52.6 | **76.5** | **84.8** | **213.9** | 1 | 32⇒16 | **59.7** | **83.7** | 88.8 | **232.2** | 1 |

Table 5: Results of text-to-video retrieval on ActivityNet Captions dataset.

| Method | Frames | R@1↑ | R@5↑ | R@10↑ | R@sum↑ | MdR↓ |
|---|---|---|---|---|---|---|
| CE (Liu et al., 2019) | - | 18.2 | 47.7 | 91.4 | 157.3 | 6 |
| ClipBERT (Lei et al., 2021) | 20 | 21.3 | 49.0 | 63.5 | 133.8 | 6 |
| Frozen (Bain et al., 2021) | 4 | 28.8 | 60.9 | - | - | 3 |
| CLIP4Clip (Luo et al., 2022) | 64 | 40.5 | 72.4 | - | - | 2 |
| Ts2net (Liu et al., 2022) | 64 | 41.0 | 73.6 | 84.5 | 199.1 | 2 |
| X-CLIP (Ma et al., 2022) | - | 46.2 | 75.5 | - | - | 6.8 |
| HBI (Jin et al., 2023) | 64 | 42.2 | 73.0 | 84.6 | 199.8 | 2 |
| BLIP† (Li et al., 2022b) | 32 | 54.3 | **80.6** | 88.8 | 223.7 | 1 |
| BLIP(Redun-A+LQ-A) | 32⇒24 | 54.6 | 80.0 | 88.8 | 223.4 | 1 |
| BLIP(Redun-A+N-InT) | 32⇒24 | **54.8** | 80.4 | **89.0** | **224.2** | 1 |

and N-InT for the text-guided frame selection. In addition, considering the competitive performance of the LQ-A, we also combine it with the Redun-A to further towards better performance of text-free frame selection. In the combinations, we calculate the cumulative scores by summing the frame scores obtained from each single frame selection method and select the top-K frames with the highest scores. In particular, when combining the Redun-A, we fix the clustering number (denoted as $Z$) referred to the retrieval performance of the single Redun-A method, that is $Z = 6$ for MSR-VTT and 16 for DiDeMo. We assign a score of $1/Z$ to each selected frame, while for the others, it is set to 0. Table 3 presents the results of the various combinations. By combining multiple frame selection methods, we can get better performance than a single frame

selection.

For example, compared with the individual Redun-A, the combination of Redun-A and LQ-A can achieve an R@sum improvement of $0.9\%$ on DiDeMo due to the elimination of the low-quality frames. When taking the text information into consideration for frame selection, the combination of Redun-A and N-InT gets the best performance at R@sum, consistently surpassing all of the single frame selection methods. We also observe that combining all of the three frame selections could not bring additional gains compared with the combinations of the two methods. We hypothesize that the N-InT could cover the ability to filter the low-quality frames to some extent. In addition, we provide the performance comparison between the combination of text-free frame selection(i.e., the combination of Redun-A and LQ-A) and the traditional key frame selection technology in the Appendix C.

### 4.3 Comparison to State-of-the-Art

To further demonstrate the effectiveness of the frame selection in TVR, we compare our method with the state-of-the-art methods in Table 4 and Table 5. It is worth noting that the pure BLIP has outperformed the current SOTA methods, which indicates that a pretrained model with a deeper fusion

module benefits the TVR. However, the accompanying drawback of the deeper fusion lies in the low efficiency. Through our frame selection, we get a better trade-off between performance and efficiency. Specifically, with proper frame selections (e.g., Redun-A + N-InT), the performance of BLIP could have a 1.0%, 3.0%, and 0.5% improvement at R@sum, while the frame number used for deeper fusion has a considerable decrease to 75%, 50% and 75% on MSR-VTT, DiDeMo and ActivityNet Captions, respectively. To verify the generalization ability of our method, we also conduct the frame selection methods on other baseline X-pool (Gorti et al., 2022) on MSR-VTT and DiDeMo datasets, as shown in Table 4. The table also clearly indicates that frame selection (e.g., Redun-A + N-InT) can effectively mitigate performance degradation despite using fewer frames, which typically results in a higher retrieval efficiency.

## 5   Conclusion

In this paper, we make an empirical study to explore the optimal frame selection methods for balancing the retrieval performance and efficiency in TVR. Beyond the existing four frame selection methods, we additionally develop two text-free paradigms, i.e., redundancy-aware and low-quality-aware frame selection. We conduct abundant experiments to analyze the impact on TVR of the above six frame selections under the condition of various frame number, and make a comprehensive comparison from the perspective of retrieval efficiency and resource consumption. Extended experimental results on multiple TVR benchmarks effectively demonstrate that with proper frame selection, we can not only get a competitive performance but also realize a higher retrieval efficiency under lower resource consumption. Hopefully, this study could spur further study on deeper interaction between text and video in TVR.

## Limitations

This empirical study provides insight into the optimal frame selections for TVR. The major limitation may be manually searching for a shared number of key frames in a dataset. However, a video typically contains a various number of key frames. Therefore, a more desirable frame selection method would be able to automatically determine an appropriate number of key frames based on the content of each video individually. We leave the related investigations to future work.

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

## A    Datasets

We introduce the three involving benchmarks for TVR as follows:

- **MSR-VTT** contains 10K videos, each of which ranges from 10 to 32 seconds and is captioned with 20 descriptions. Each description provides a brief overview of the corresponding video for standard sentence-to-video retrieval. We follow the widely-used data split in previous methods (Gabeur et al., 2020; Luo et al., 2022) to train the model on the training-9K set and test on the test 1k-A set.

- **DiDeMo** includes 10K unedited videos paired with 40K descriptions. Each video is divided into multiple segments, and each segment is accompanied by a corresponding description. Following previous works (Lei et al., 2021; Luo et al., 2022), we conduct paragraph-to-video retrieval by concatenating all descriptions of each video into a single query.

- **ActivityNet Captions** contains 20K videos annotated with 100K descriptions. The training set contains 10K videos, and we use the val1 set with 4.9K videos to report results. Following (Gabeur et al., 2020; Luo et al., 2022), all descriptions of a video are also concatenated together for paragraph-to-video retrieval.

## B    Implementation Details.

All experiments are conducted on 4 NVIDIA A40 GPUs. We initialize the model parameters used for empirical study from the BLIP (Li et al., 2022b)

Table 6: Performance comparison between ours Redun-A+LQ-A and Katna on DiDeMo. Both methods are designed to eliminate redundant and low-quality frames.

| Method | Frames | R@1↑ | R@5↑ | R@10↑ | R@sum↑ | MdR↓ |
|---|---|---|---|---|---|---|
| Katna (KeplerLab, 2019) | 8 | 52.3 | 77.8 | 85.7 | 215.8 | 1 |
| | 16 | 56.0 | 81.0 | 86.6 | 223.6 | 1 |
| Redun-A+LQ-A | 8 | 57.9 | 82.3 | 88.3 | 228.5 | 1 |
| | 16 | 59.5 | 83.5 | 89.1 | 232.1 | 1 |

that has been fine-tuned on the COCO (Lin et al., 2014). As for the extra parameters of the frame selection module of the LQ-A and InT, we use Kaiming Normal (He et al., 2015) initialization. To convert a video into a frame sequence, we uniformly pre-sample 16, 32, and 32 frames from each video of MSR-VTT, DiDeMo, and ActivityNet Captions, respectively, and then resize each frame to 224×224. The maximum textual length is set as 35 for MSR-VTT and 64 for the other two datasets. We set the batch size to 4 and set the learning rate for BLIP-initialized weights to 5e-6 and for other parameters to 5e-4. We optimize our model for 5 epochs using the AdamW optimizer (Loshchilov and Hutter, 2017) with a weight decay of 0.05 and decay the learning rate using a cosine schedule.

Except for the BLIP, we also employ another baseline X-pool (Gorti et al., 2022) to verify the effectiveness of the optimal frame selection methods in Section 4.3. We follow the default settings of the X-pool for MSR-VTT. For the DiDeMO, we set the batch size to 16 and the pre-sampled frames to 32. For the scorer network of LQ-A, we set its learning rate to 5e-4.

## C    Comparison to Traditional Key Frame Selection Method

Key frame extraction techniques (Chang et al., 1999; Divakaran et al., 2002; Chang, 2003) have been developed in the past decades. However, these techniques are generally based on the low-level pixel-level understandings of videos (e.g. frames' color feature) for the key frame selection. For instance, Katna (KeplerLab, 2019) identifies redundant and blurry frames by initially applying K-means Clustering to frames using image histograms. It subsequently selects the best frames from clusters based on the variance of the Laplacian and the overall sharpness of the frames. In Table 6, we compare the retrieval performance achieved by applying Redun-A+LQ-A and Katna for frame selection, respectively. Both methods are designed

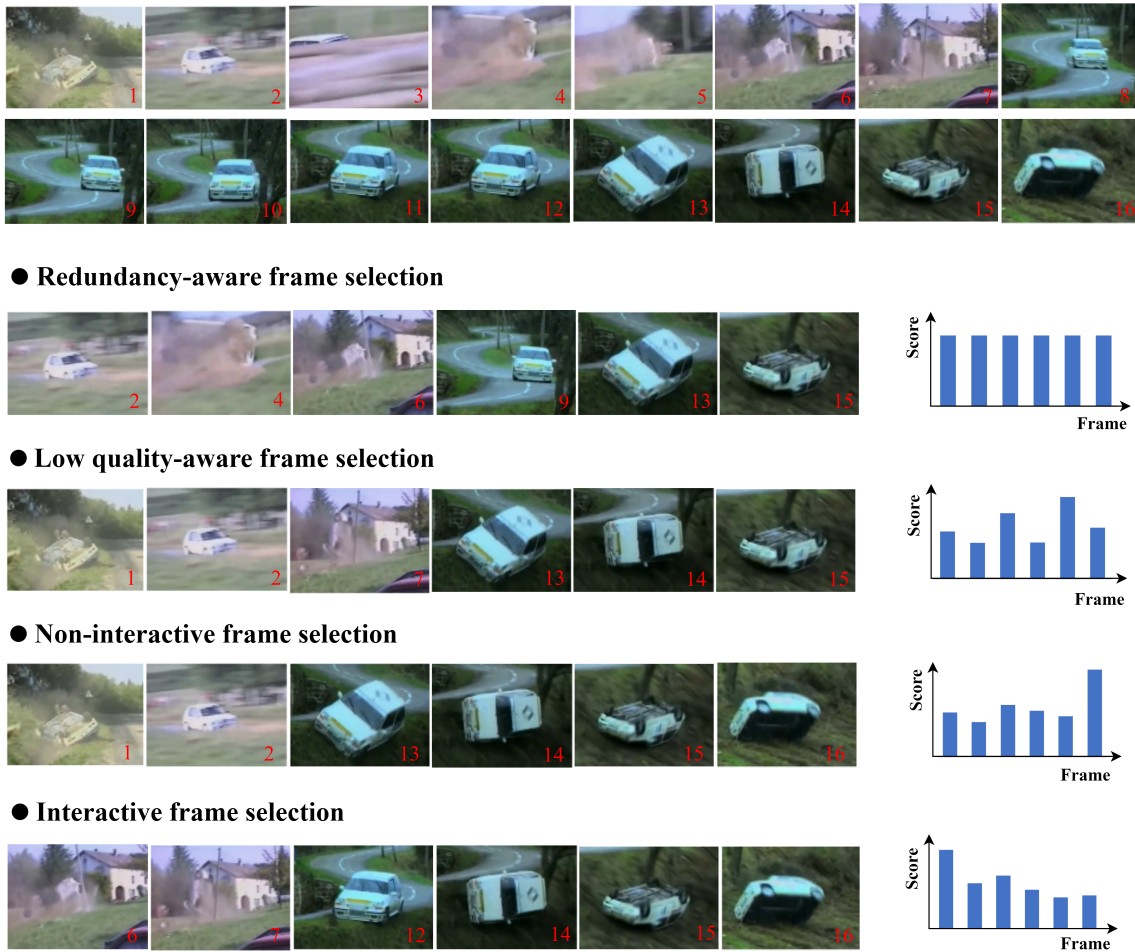

Figure 2: Visualization of the selected frames under different frame selection methods. We select 6 key frames with the highest scores in the raw pre-sampled frames. Bar plots show the corresponding scores of the displayed frames.

to eliminate redundant and low-quality frames. Our Redun-A+LQ-A exhibits a significant advantage in TVR compared to Katna, with performance improvements of 3.5% and 5.6% at R@1 when the frame number is set to 16 and 8, respectively. Our Redun-A+LQ-A select frames using frame features computed by the video encoder, which encapsulate the understanding of video contents. In comparison, the Katna pre-processes videos based on color features. On one hand, Katna fails to prioritize the primary area of interest in frames during the selection process. On the other hand, the combination of the Katna and deep retrieval model could only be done by video pre-processing. Therefore, our frame selection methods are more friendly to the deep retrieval model and thus get better performance.

## D    Visualization of Frame Selection

We display visualization results under various frame selection methods in Figure 2. We can see that the selected frames could effectively reflect the purpose of each selection method. Specifically, the redundancy-aware frame selection retains the most diverse information of the pre-sampled frames, while the low-quality-aware frame selection effectively recognizes the blurry frames (i.e., the $2^{nd} \sim 4^{th}$ frames). In contrast, the text-guided frame selection, including the non-interactive and the interactive frame selection, can retain almost all the frames that are related to the query text. The visualization illustrates that the involved frame selection methods could effectively filter the inessential frames based on the respective criteria.