# OpenReview forum: "An Empirical Study of Frame Selection for Text-to-Video Retrieval"
_EMNLP/2023/Conference — EMNLP 2023 Findings_

### Official Review · Reviewer_tb4G · 2023-08-01

**Soundness:** 3

**Excitement:**

3: Ambivalent: It has merits (e.g., it reports state-of-the-art results, the idea is nice), but there are key weaknesses (e.g., it describes incremental work), and it can significantly benefit from another round of revision. However, I won't object to accepting it if my co-reviewers champion it.

**Paper Topic And Main Contributions:**

The paper made an empirical study of frame selection for Text-To-Video Retrieval, and systemically classified existing frame selection methods into text-free and text-guided ones. The paper developed two frame selection methods, i.e. Redundancy-Aware frame selection and Low Quality-Aware frame selection. The paper conducted a number of experiments to verify the impact of frame selection methods on the efficiency and retrieval performance of TVR tasks. The experiments show that proper frame selections can improve retrieval efficiency without sacrificing the retrieval performance of TVR.

**Questions For The Authors:**

There are some writing errors in the paper as follows:
1. Line 240:  F1 type error
2. Figure 1(b): The index after frame selection should range from 1 to K. Only (I & II) meet this requirement.

**Reasons To Accept:**

1. The Redundancy-Aware frame selection utilizes the k-medoids++ clustering algorithm to partition the representations of each frame into K groups. Choose the representation of the center frame of each group on behalf of the whole group. This eliminates redundant frames.
2. The Low Quality-Aware frame selection appends a score network after the vision encoder to assess the quality score of each frame. Leverage the paired text as weak supervision to optimize the score network. This eliminates the low-quality frames.
3. The six frame selection methods are verified in more detailed experiments, and the two proposed frame selection methods improve the efficiency of the TVR task without sacrificing the retrieval performance.

**Reasons To Reject:**

1.The evaluation indicators are not very reasonable. R@sum is rarely seen in articles in TVR.
2.The experiment part uses too much space on the influence of different frame selections on the retrieval performance, which has little improvement. However, it spends little space on the effect of different frame selections on the retrieval efficiency, whose improvement is very obvious.
3.In experiments, when combining different frame selections, the paper does not show how the hyper parameter K affects the results, which is similar to the analyses in Table 4. Drawing the conclusion that proper frame selections improve the retrieval efficiency without sacrificing the retrieval performance of TVR,  should be more rigorous.

**Reproducibility:**

4: Could mostly reproduce the results, but there may be some variation because of sample variance or minor variations in their interpretation of the protocol or method.

**Reviewer Confidence:**

4: Quite sure. I tried to check the important points carefully. It's unlikely, though conceivable, that I missed something that should affect my ratings.

---

> ### Author Rebuttal · Authors · 2023-08-28
>
> Thanks for your constructive comments and suggestions, and they are exceedingly helpful for us to improve our paper. In the following, the comments are first stated and then followed by our point-to-point responses.
>
>
>
> > Q1: The evaluation indicators are not very reasonable. R@sum is rarely seen in articles in TVR.
>
> Thank you for pointing out the problem of using the uncommon evaluation indicator R@sum. As the R@sum could reflect the overall performance of R@1, R@5 and R@10, we only display the R@1 and R@sum in Table 1 and Table 3 in the main paper in the condition of space limitation. We will add the results of all metrics(R@1, R@5, R@10 and MdR) in the Appendix of the revised version.
>
> > Q2: The experiment part uses too much space on the influence of different frame selections on the retrieval performance, which has little improvement. However, it spends little space on the effect of different frame selections on the retrieval efficiency, whose improvement is very obvious.
>
> Thanks for your positive comments on the effect of our frame selection for efficient TVR.
>
> Considering that the retrieval efficiency is explicitly related to the frame number (more frames mean lower efficiency and vice versa), we thus only present the efficiency comparison under **the optimal setting of frame number** for each method in Table 2 in our manuscript, i.e., comparing the efficiency of the methods with their optimal performance. The prerequisite for it is the detailed analysis of the influence of different frame selections on the retrieval performance in Table 1 in our manuscript.
>
> Nevertheless, we also add the analysis of the retrieval efficiency on MSR-VTT datasets, as shown in the table below.
>
> | Method  | Frame | R@1  | MeM(GB) | GLOPs  | Time(ms) |
> | :-----: | :---: | :--: | :-----: | :----: | :------: |
> |  Base   |  16   | 52.9 |  19.2   | 1107.6 |  245.8   |
> |   Uni   |  12   | 51.7 |  15.8   | 836.7  |  191.9   |
> |  Rand   |  12   | 51.8 |  15.8   | 836.7  |  191.9   |
> | Redun-A |   6   | 52.2 |  14.9   | 861.7  |  112.1   |
> |  LQ-A   |  12   | 51.1 |  17.6   | 1042.2 |  189.8   |
> |  N-InT  |  12   | 50.4 |  17.6   | 1070.1 |  197.2   |
> |   InT   |  12   | 50.8 |  17.6   | 1098.6 |  210.6   |
>
> \*  Efficiency comparison of frame selection method under the best performance on **MSRVTT**.
>
> We speed up the inference by the frame selection, as shown in the last column. The Uni and Rand determine the K keyframe before the vision encoder, resulting in the lowest MeM and GLOPs. The Redun-A retains the least keyframes and thus the lowest MeM. Compared to the text-free methods, the text-guided methods are less efficient. We will add the experimental results on MSRVTT in the revised paper.
>
> > Q3: 1. In experiments, when combining different frame selections, the paper does not show how the hyper parameter K affects the results, which is similar to the analyses in Table 4.
>
> We appreciate that you point out that we ignore the more detailed discussion about the influence of the hyperparameter K in the combination of frame selection methods. We thus append the experiments with different hyperparameters of K, shown in the below tables.
>
> | Frame |   R@1    |   R@5    |   R@10   |   R@sum   | MdR  |
> | :---: | :------: | :------: | :------: | :-------: | :--: |
> |  24   |   59.4   | **83.7** |   88.9   |   232.0   |  1   |
> |  16   | **59.5** |   83.5   | **89.1** | **232.1** |  1   |
> |  12   |   57.6   |   83.7   |   89.3   |   230.6   |  1   |
> |   8   |   57.9   |   82.3   |   88.3   |   228.5   |  1   |
>
> \*  Retrieval performance of the combination of **Redun-A and LQ-A** under the different setting of $K$ on **DiDeMo**.
>
> | Frame |   R@1    |   R@5    |   R@10   |   R@sum   | MdR  |
> | :---: | :------: | :------: | :------: | :-------: | :--: |
> |  24   |   59.3   |   82.6   |   87.3   |   229.2   |  1   |
> |  16   | **59.7** | **83.7** | **88.8** | **232.2** |  1   |
> |  12   |   59.0   |   83.2   |   88.4   |   230.6   |  1   |
> |   8   |   55.8   |   82.0   |   87.6   |   225.4   |  1   |
>
> \*  Retrieval performance of the combination of **Redun-A and N-InT** under the different setting of $K$ on **DiDeMo**.
>
> | Frame |   R@1    |   R@5    |   R@10   |   R@sum   | MdR  |
> | :---: | :------: | :------: | :------: | :-------: | :--: |
> |  24   | **60.1** |   83.6   |   88.9   | **232.6** |  1   |
> |  16   |   59.2   | **83.9** | **89.2** |   232.3   |  1   |
> |  12   |   57.7   |   83.9   |   89.1   |   230.7   |  1   |
> |   8   |   58.4   |   83.8   |   88.9   |   231.1   |  1   |
>
> \*  Retrieval performance of the combination of **Redun-A, LQ-A and N-InT** under the different setting of $K$ on **DiDeMo**.
>
> For DiDeMo, the optimal values of K that lead to the highest performance are 16, 16, and 24 for the three different cases, respectively. For the case of Redun-A+LQ-A+N-InT on DiDeMo, when K is set to 16, the achieved R@sum remains comparable to the results obtained with K=24 while with higher retrieval efficiency. Consequently, we opt to utilize K=16 for the comparison between single frame selection and the combinations of frame selection, as presented in Table 3 of our manuscript. Except for the shown results of DiDeMo, We will append all the experiments MSRVTT in the revised Appendix.
>
> > Q3: 2.  Drawing the conclusion that proper frame selections improve the retrieval efficiency without sacrificing the retrieval performance of TVR, should be more rigorous.
>
> Thanks for your constructive comment. We agree that the expression of "proper" frame selection is not rigorous enough. We have concluded the exact optimal frame selection methods in Lines 115~122 of our manuscript, that is, the combination of Redun-A and LQ-A and the combination of Redun-A and N-InT. According to the results in Table 2 of our manuscript, the combination of Redun-A and N-InT performs slightly better than the combination of Redun-A and LQ-A, thanks to the text guidance. However, as shown in Table 3 of our manuscript, LQ-A has slightly higher efficiency than N-Int owing to the text-free frame selection. Overall, both combinations are simple but effective. We will modify the proper frame selection to the exact frame selection methods in the revised version.
>
> > Q4:  There are some writing errors in the paper as follows.
> >
> > 1. Line 240: F1 type error
> > 2. Figure 1(b): The index after frame selection should range from 1 to K. Only (I & II) meet this requirement.
>
> Thanks for your careful comments. We will correct the mistakes and carefully check other possible errors in the revised vision.

---

### Official Review · Reviewer_FXRQ · 2023-08-06

**Soundness:** 3

**Excitement:**

4: Strong: This paper deepens the understanding of some phenomenon or lowers the barriers to an existing research direction.

**Paper Topic And Main Contributions:**

This paper proposed two simple approaches to sample frames for efficient text-to-video retrieval while remaining performance. They also compare the performance to existing baselines and demonstrate that the proposed method can achieve satisfactory performance.

**Reasons To Accept:**

- The authors proposed two simple yet effective text-free methods to improve the frame selection of text-to-video retrieval
- Experimental results show that the combination of two proposed text-free methods can outperform other text-free baselines. In the scenario of text-guided methods, the proposed redundant-aware method can also outperform the baselines.

**Reasons To Reject:**

I have some concerns about the following points:
- in the training of the LQ-A module, is the text paired with a single frame or the whole video clip? How to guarantee that the similarity can reflect the quality of a frame if the text is paired with a video clip? A partially matched frame could also play an important role in a video clip. How about considering the explicit lens movement, transition, and blurred frames, which are easy to detect, making the system interpretable?
-  How the LQ-A and Redun-A are combined? The calculation of the score of Redun-A is not clear.
- The overall performance of text-free methods is better than the text-guided methods which involve additional query information and higher computational cost. If it's properly implemented, could you please provide more analysis about it?

**Reproducibility:**

4: Could mostly reproduce the results, but there may be some variation because of sample variance or minor variations in their interpretation of the protocol or method.

**Reviewer Confidence:**

3: Pretty sure, but there's a chance I missed something. Although I have a good feel for this area in general, I did not carefully check the paper's details, e.g., the math, experimental design, or novelty.

---

> ### Author Rebuttal · Authors · 2023-08-28
>
> We deeply appreciate your thoughtful comments in reviewing our manuscript. Our point-to-point responses to your comments are given below.
>
>
>
> > Q1: 1. in the training of the LQ-A module, is the text paired with a single frame or the whole video clip.
>
> In the datasets of TVR, each text is paired with a video clip. Therefore, in the training of the LQ-A module, we constrain the text paired with the whole video clip by the VTM loss, formulated in Equal (8) ~ (10) in our manuscript.
>
> > Q1: 2. How to guarantee that the similarity can reflect the quality of a frame if the text is paired with a video clip?
>
> The quality scores of the frames calculated by the LQ-A module are not in line with the similarity with the paired texts. The LQ-A evaluates the quality of the frames from the perspective of the video content itself rather than relying on the similarity to the particular text. During training, we first score each frame by the scorer network (formulated in Equal (3) in our manuscript) and then conduct a weighted mean based on the score of each frame in a video (formulated in Equal (2) in our manuscript). Finally, by the VTM loss, we optimize the scorer network of LQ-A.
>
> What's more, we provide more experiments to validate that the LQ-A's frame scores could reflect the frames' quality after training with texts. The principle is that **retrieval with low-quality frames performs worse**. We select high-quality frames with the highest K scores (denoted as top-K ) and low-quality frames with the lowest K scores (denoted as last-K) during inference. Subsequently, we proceed to compare the retrieval performance between these two sets. As shown in the tables below, the retrieval performance of high-quality frames exhibits an improvement of over 20% in terms of R@sum when contrasted with the performance of the low-quality frames. And with the frame number decreasing, the advantage of retrieval with the high-quality frame is more obvious. These imply that the discarded frames are of low quality and harmful for retrieval.
>
> | Frame  | R@1  | R@5  | R@10 | R@sum | MdR  |
> | :----: | :--: | :--: | :--: | :---: | :--: |
> | top-8  | 54.5 | 80.5 | 87.5 | 222.5 |  1   |
> | last-8 | 39.6 | 66.6 | 77.1 | 183.3 |  2   |
> | top-4  | 51.7 | 76.8 | 85.0 | 213.5 |  1   |
> | last-4 | 34.2 | 59.7 | 71.0 | 164.9 |  3   |
> | top-1  | 36.3 | 63.5 | 73.4 | 173.2 |  3   |
> | last-1 | 23.6 | 45.5 | 57.2 | 123.6 |  7   |
>
> \*  Comparisions of retrieval with high-quality frames and low-quality frames on **DiDeMo**.
>
> > Q1: 3.  A partially matched frame could also play an important role in a video clip.
>
> Thanks for your insightful comments. We agree that a partially matched frame could also play an important role in a video clip. However, the partially matched frames may contribute less to the correct retrieval. We could get a better trade-off between effectiveness and efficiency by discarding these frames with proper frame selection.
>
> > Q1: 4.  How about considering the explicit lens movement, transition, and blurred frames, which are easy to detect, making the system interpretable?
>
> We agree with the review that using video quality detection technology could make the system more interpretable. However, the final goal of the frame selection is to construct a **highly effective and efficient** retrieval system.
>
> From the efficiency perspective, our LQ-A offers an **end-to-end** video information processing method and could be integrated into the deep retrieval model conveniently. It incorporates a single multilayer perceptron for frame scoring, ensuring comparable efficiency to Katna.
>
> In terms of effectiveness, we evaluate the retrieval performance of LQ-A compared to Katna[1], a method that offers video blur detection. As demonstrated in the table below, LQ-A outperforms Katna, potentially due to its ability to comprehend video content and make informed assessments.
>
> | Method | Frame | R@1  | R@5  | R@10 | R@sum | MdR  |
> | :----: | :---: | :--: | :--: | :--: | :---: | ---- |
> | Katna  |  16   | 56.0 | 81.0 | 86.6 | 223.6 | 1    |
> |  LQ-A  |  16   | 57.7 | 83.0 | 88.0 | 228.7 | 1    |
> | Katna  |   8   | 52.3 | 77.8 | 85.7 | 215.8 | 1    |
> |  LQ-A  |   8   | 55.9 | 81.2 | 87.4 | 224.5 | 1    |
>
> \*  Comparison between retrieval with frames extract by the Katna and ours on **DiDeMo**.
>
> Therefore, our LQ-A is more reasonable for TVR from both perspectives of effectiveness and efficiency.
>
> [1] https://github.com/keplerlab/katna
>
> > Q2: How the LQ-A and Redun-A are combined? The calculation of the score of Redun-A is not clear.
>
> The combination of the LQ-A and Redun-A is described in Lines 546 to 549 in the manuscript. We calculate the frame scores by LQ-A and Redun-A, respectively, and then sum the scores of corresponding frames. Finally, we select top-K frames with high scores. For the Redun-A, we fix the clustering number (denoted as $Z$ ) referred to the retrieval performance of the single Redun-A method. We assign a score of $1/Z$ to each selected frame, while for the others, it is 0. Thanks to the reviewer for reminding us of the important details of the score calculation of Redun-A, and we will append it in the revised version.
>
> > Q3: The overall performance of text-free methods is better than the text-guided methods which involve additional query information and higher computational cost. If it's properly implemented, could you please provide more analysis about it?
>
> Thanks for your question. By our analysis, we conjecture that the text-guided frame selection introduces the **text bias** for TVR (described in Lines 503~510 in our manuscript). The video content encompasses a wide range of diversity. **Selecting frames according to their similarities with the given text may improve the similarities of the videos partially related to the given text, leading to extra disturbance for TVR.** We show the results of conducting N-InT during inference for a clearer understanding. The results are shown in the table below.
>
> | Training + Inference | R@1  | R@5  | R@10 | R@sum | MdR  |
> | :------------------: | :--: | :--: | :--: | :---: | :--: |
> |      Uni + Uni       | 51.7 | 76.1 | 84.8 | 212.6 |  1   |
> |     Uni + N-InT      | 50.4 | 75.5 | 84.4 | 210.3 |  1   |
>
> \*  Comparision between inference with Uni and N-InT based $K=12$ on **MSRVTT**.
>
> From the table, the inference with N-InT leads to a decrease in performance, which verifies the text bias indeed destroys the performance of TVR. We will append more analysis and visualization in the revised version.

---

### Official Review · Reviewer_ELKD · 2023-08-06

**Typos Grammar Style And Presentation Improvements:** 1. Line 092 please replace "there lac…
**Soundness:** 4

**Excitement:**

4: Strong: This paper deepens the understanding of some phenomenon or lowers the barriers to an existing research direction.

**Missing References:**

https://www.semanticscholar.org/paper/Motion-activity-based-extraction-of-key-frames-from-Divakaran-Peker/515f899f61c33782ec78045fa849e27e607d9985

https://www.ee.columbia.edu/~sfchang/papers/talk-iciap-0903.pdf

https://ieeexplore.ieee.org/document/809161

Please see comments above to respond to these citations.

**Paper Topic And Main Contributions:**

This paper presents two new video keyframe extraction techniques, the first based on redundancy reduction and the second based on quality detection, which are complementary approaches to keyframe extraction. The paper presents a thorough evaluation of state of the art techniques for key frame extraction on standard video retrieval datasets. The authors demonstrate clear advancement of the state of the art using their proposed novel keyframe extraction techniques.

**Questions For The Authors:**

Have you considered developing intuitive explanations for why your proposed techniques work better than the state of the art?
What is the novelty of your redundancy aware approach over Zhao et al (whom you cite) given that you seem to be doing exactly what they did?

**Reasons To Accept:**

1. Thorough review of the state of the art of post-deep learning key frame extraction methods.
2. Clear improvement over the state of the art.
3. Intuititively satisfying results.
4. Well written paper.

**Reasons To Reject:**

1, This is not a weakness peculiar to these authors. In general the video summarization literature seems to show ignorance of video summarization techniques developed in the multimedia community more than 20 years ago. Those were techniques that did not rely on the elaborate machine learning techniques on display at this moment but worked quite well. Here are three examples of such papers:

https://www.semanticscholar.org/paper/Motion-activity-based-extraction-of-key-frames-from-Divakaran-Peker/515f899f61c33782ec78045fa849e27e607d9985

https://www.ee.columbia.edu/~sfchang/papers/talk-iciap-0903.pdf

https://ieeexplore.ieee.org/document/809161

An inspection of the bibliographies of these papers will reveal that there have been a number of keyframe extraction techniques developed in the past. Would be nice of the authors to address why their technique establishes novelty over those methods.
Note that I don't want to single these authors out since the entire vision community seems to be reinventing past techniques while also genuinely exploiting the power of deep learning techniques that did not exist in the times of the three papers cited above.

**Reproducibility:**

5: Could easily reproduce the results.

**Reviewer Confidence:**

5: Positive that my evaluation is correct. I read the paper very carefully and I am very familiar with related work.

---

> ### Author Rebuttal · Authors · 2023-08-29
>
> Thanks for your constructive comments and suggestions, which are exceedingly helpful for us to improve our manuscript. Our point-to-point responses to your comments are given below.
>
> In the following, your comments are first stated and then followed by our point-to-point responses.
>
>
>
> >Q1:An inspection of the bibliographies of these papers will reveal that there have been a number of keyframe extraction techniques developed in the past. Would be nice of the authors to address why their technique establishes novelty over those methods.
>
> Thanks for your insightful comments and constructive suggestions. The techniques have shown charmful performance in the multimedia community in the past decades and could be one of the reasonable methods for frame selection nowadays. However, the past techniques[1,2,3] are generally based on the low-level pixel-level understandings of videos (e.g. frames' color feature) for the keyframe selection. In comparison, our frame selection methods enable the selection of frames grounded in **high-level semantic comprehension of videos**, and can be naturally integrated into the deep retrieval model to empower an **end-to-end** learning.
>
> In addition, to further show the effect of the semantic-level understanding of videos for frame selection in TVR, we apply the existing keyframe extraction technology[4] (called *Katna*) for TVR. Initially, it employs K-means Clustering on frames using image histograms to identify **redundant frames**. Subsequently, from the optimal frame within these clusters, Katna detects **image blur**. This detection is rooted in the analysis of both the variance of Laplacian and the frame's overall sharpness. The results are shown below.
>
> |    Method    | Frame | R@1  | R@5  | R@10 | R@sum | MdR  |
> | :----------: | :---: | :--: | :--: | :--: | :---: | ---- |
> |    Katna     |  16   | 56.0 | 81.0 | 86.6 | 223.6 | 1    |
> | Redun-A+LQ-A |  16   | 59.5 | 83.5 | 89.1 | 232.1 | 1    |
> |    Katna     |   8   | 52.3 | 77.8 | 85.7 | 215.8 | 1    |
> | Redun-A+LQ-A |   8   | 57.9 | 82.3 | 88.3 | 228.5 | 1    |
>
> \*  Comparison between retrieval with frames extract by the Katna and ours on **DiDeMo**.
>
> Our Redun-A+LQ-A surpass the retrieval performance of Katna significantly. In comparison, our Redun-A and LQ-A select frames using frame features computed by the video encoder. These features encapsulate the comprehension of video content. However, the Katna pre-processes videos based on color features. On one hand, the combination of the Katna and deep retrieval model could only be done by video pre-processing. On the other hand, the method fails to prioritize the primary area of interest during the selection process. Therefore, our frame selection methods are more friendly to the deep retrieval model and thus get better performance.
>
> We will add bibliographies of past video summarization technologies and clarify the difference of our deep frame selection compared with these technologies in the revised version. The experimental comparison of ours with Katna will also be appended in the revised version.
>
> [1] Chang H S, Sull S, Lee S U. Efficient video indexing scheme for content-based retrieval[J]. IEEE Transactions on Circuits and Systems for Video Technology, 1999, 9(8): 1269-1279.
>
> [2] Chang S F. Content-based video summarization and adaptation for ubiquitous media access[C]//12th International Conference on Image Analysis and Processing, 2003. Proceedings. IEEE, 2003: 494-496.
>
> [3] Divakaran A, Radhakrishnan R, Peker K A. Motion activity-based extraction of key-frames from video shots[C]//Proceedings. International Conference on Image Processing. IEEE, 2002, 1: I-I.
>
> [4] https://github.com/keplerlab/katna
>
> > Q2: 1. Have you considered developing intuitive explanations for why your proposed techniques work better than the state of the art?
>
> Thanks for your question. Our techniques are designed for detecting **redundant and noisy frames** within videos for TVR. On the one hand, the retrieval with redundant frames leads to unnecessary computational load and time consumption. On the other hand, noise frames, such as those of low-quality and text-irrelated frames, not only bring unnecessary resource expenditure but also disrupt the alignment between texts and their associated videos. We **simultaneously** address both scenarios by empirically combining multiple frame selection methods to refine video frames. We thus achieve both satisfying effectiveness and efficient retrieval outcomes.
>
> We will provide more intuitive explanations for the effectiveness of our methods in the revised version.
>
>
>
> > Q2: 2. What is the novelty of your redundancy aware approach over Zhao et al (whom you cite) given that you seem to be doing exactly what they did?
>
> Thanks for your question. Compared to the CenterCLIP[5], we propose a new strategy, **frame-level frame selection**, for solving video redundancy. On the one hand, CenterCLIP operates at the patch token-level features to identify video redundancy, whereas our method operates at the frame-level features, which contributes to more efficient information selection. On the other hand, CenterCLIP focuses on constraint redundancy within consecutive frames, whereas our frame-level frame selection approach reduces redundancy across the entirety of the video contents. By the frame-level frame selection, our proposed selection strategy gains both higher efficiency and performance.
>
> For the retrieval performance, we conduct the token-level information selection (CenterCLIP) in our framework. We ensure the number of selected tokens in CenterCLIP corresponds to the same informational quantity as the number of frames in the Redun-A approach. The results are shown in the table below.
>
>
> |   Method   | R@1  | R@5  | R@10 | R@sum | MdR  |
> | :--------: | :--: | :--: | :--: | :---: | :--: |
> | CenterCLIP | 59.7 | 81.6 | 87.3 | 228.6 |  1   |
> |  Redun-A   | 59.3 | 83.3 | 88.5 | 231.1 |  1   |
>
> \*  Comparision between CenterCLIP with Redun-A based $K=16$ on **DiDeMo**.
>
> Our frame selection achieves better performance than CenterCLIP overall. Our Redun-A **retains the integrity of the spatial information**, which is benifit to the learning of frame-level representation and further video representation, thus benefiting the TVR.
>
> For the selection efficiency, the CenterCLIP clusters abundant video patch tokens (**N*P, P>>N**), while our Redun-A clusters among the frame-level tokens (**N**) in each video. As a result, in the information selection process, our Redun-A greatly saves computational resources and time consumption and thus exhibits a more efficient way for information selection.
>
> We will append the explanations of the novelty and advantages of Redun-A compared to the CenterCLIP in the revised version.
>
> [5] Zhao S, Zhu L, Wang X, et al. Centerclip: Token clustering for efficient text-video retrieval[C]//Proceedings of the 45th International ACM SIGIR Conference on Research and Development in Information Retrieval. 2022: 970-981.
>
> > Q3. Line 092 please replace "there lacks" with " the state of the art lacks" ; Line 568 please replace "State of the arts" with "State of the art"
>
> Thanks for your comments. We will correct the two minor mistakes and carefully check other possible errors in the revised version.

---

### Meta-Review · Area_Chair_4aus · 2023-09-17

**Recommendation:** 3

**Metareview:**

This paper presents two new methods for frame selection for Text-to-Video Retrieval (TVR). The paper has some merits. For example, reviewers agreed the proposed methods are effective as they show gains over baselines, and some reviewers enjoyed the clarity of the paper. However, reviewers also raised some concerns on the experimental settings and incremental novelty. Overall, AC agrees that the paper is a good source of experimental findings and methods of improvements in TVR.

---

### Decision · Program_Chairs · 2023-10-07

**Decision:**

Accept-Findings

**Comment:**

This paper presents two new methods for frame selection for Text-to-Video Retrieval (TVR). The paper has some merits. For example, reviewers agreed the proposed methods are effective as they show gains over baselines, and some reviewers enjoyed the clarity of the paper. However, reviewers also raised some concerns on the experimental settings and incremental novelty. Overall, AC agrees that the paper is a good source of experimental findings and methods of improvements in TVR.